# Taking a Pulse on Community Participation in Maternal Health through Community Clinics in Bangladesh

**DOI:** 10.3390/ijerph20032271

**Published:** 2023-01-27

**Authors:** Goutom Banik, Tapas Mazumder, Abu Bakkar Siddique, A.F.M Azim Uddin, Shams El Arifeen, Janet Perkins, Ahmed Ehsanur Rahman

**Affiliations:** 1Save the Children, Dhaka 1212, Bangladesh; 2Health Research Institute, Faculty of Health, University of Canberra, Canberra, ACT 2617, Australia; 3Maternal and Child Health Division, International Centre for Diarrhoeal Disease Research, Bangladesh (icddr,b), Mohakhali, Dhaka 1212, Bangladesh; 4Department of Social Anthropology, School of Social & Political Science, University of Edinburgh, 15a George Square, Edinburgh EH8 9LD, Scotland, UK

**Keywords:** community participation, community clinics, functionality, community group, community support group, maternal health

## Abstract

Bangladesh started institutionalising community participation by setting-up community clinics (CCs) during the mid-90 s. This paper presents the genealogy of CCs, the community participation mechanism embedded within CCs, and the case of 54 CCs in Brahmanbaria, through the lens of maternal health. We undertook a desk review to understand the journey of CCs. In 2018, we assessed the accessibility, readiness and functionality of CCs, and a household survey to know recently delivered women’s perceptions of CC’s community groups (CGs) and community support groups (CSGs). We performed multiple logistic regression to determine the association between the functionality of these groups and women’s perception regarding these groups’ activities on maternal health. The integration of community participation involving CCs started to roll out through the operationalisation of the Health and Population Sector Programme 1998–2003. In 2019, 13,907 CCs were operational. However, per our CC assessment, their accessibility and readiness were moderate but there were gaps in the functionality of the CCs. The perception of women regarding these groups’ functionality was significantly better when the group members met regularly. The gaps in CCs are primarily induced by the shortcomings of its community participation model. Proper understanding is needed to address this problem which has many facets and layers, including political priorities, expectations, and provisions at a local level.

## 1. Background

The International Conference on Primary Health Care (PHC), held in Alma-Ata in 1978, marked a monumental moment in global health discourses. During this conference, and enshrined in the declaration that followed it, the 134 participating countries committed to protecting and promoting ‘Health for All’ and expanding their vision beyond the health sector to emphasize other areas which influence health. These include social and economic development, promotion of equity, the participation of individuals, families, and communities in planning and implementing health care, developing and implementing national strategies for sustaining PHC, and inter-country coordination for ensuring PHC [1]. Since then, community participation has been globally recognized as one of the critical components for improving PHC to safeguard ‘Health for All’ [1,2,3,4,5]. However, promoting community participation proved challenging to implement, and many countries have continued to emphasize disease-focused vertical interventions prevalent in global health instead of prioritizing PHC [6,7].

The primary mechanism adopted in Bangladesh to institutionalise community participation is establishing community clinics (CCs). Through the five-year plan of 1998–2003, the government aimed to provide an essential service package (ESP) to vulnerable groups of society. Their strategies involved reforming the service delivery strategy and moving from house-to-house visits to a fixed-structure service delivery mechanism [8]. Policymakers designed CCs to bring facility-based essential health services closer to the people in a community. As per the national guideline, every CC is to be managed by a community group (CG) and supported by three community support groups (CSGs). These CGs and CSGs are responsible for maximizing community participation in the public health sector and giving them a voice in the design, delivery, and monitoring of health services [9,10]. However, despite over two decades of implementation and popularity among development partners, not much is known regarding the functionality of CCs and community participation embedded within the model. 

This paper presents a genealogy of the CCs in Bangladesh, specifically the community participation mechanism embedded within the CCs, to contextualize the institutionalization of formal mechanisms to promote community participation in health in the country. It then presents the case of two upazilas (sub-districts) of Brahmanbaria to take a pulse on the readiness and accessibility of the CCs, the functionality of CGs and CSGs, and women’s perceptions regarding community participation in maternal health care through these platforms. This paper adds to the existing knowledge on the use of community participation for improving maternal health. The new knowledge can be used to further improve the community participation model for achieving better maternal health outcomes. 

## 2. Methods

### 2.1. Documentation of Genealogy

We performed a desk review of 27 documents to understand the journey of CCs in Bangladesh. These documents were identified through a web-based search (Google and Google Scholar search), consultation with researchers, programme implementers, and policymakers. The documents include policy documents, peer-reviewed research papers, theses and dissertations, election manifestos of political parties, and websites of international organisations. 

### 2.2. Health Facility Assessment

We conducted health facility assessments using a structured checklist between July and September 2017 to determine service accessibility, readiness, and functionality of CCs in two upazilas of the Brahmanbaria district in Bangladesh. The assessment tool was adapted from the Bangladesh Health Facility Survey (BHFS)-2014 tool and also based on the review of the 4th Health, Population and Nutrition Sector Programme (HPNSP) [11,12]. We recruited and trained two assessors who had more than five years of experience working with the Bangladesh health systems. We covered all CCs of the selected upazilas, i.e., 21 CCs in Sarail and 33 CCs in Kasba. However, the infrastructure of one CC at Kasba and another at Sarail was destroyed. Furthermore, despite multiple visits, we could not collect data from four CCs of Kasba due to the unavailability of service providers during our visits.

**Accessibility to CCs:** We assessed the accessibility status based on the condition of the access road to the CCs in both dry and rainy seasons, as reported by CC-based healthcare providers. 

**Readiness of CCs:** We generated the readiness score for CCs based on ten essential items, which were finalised based on the Bangladesh Health Facility Survey 2014 tool, Operational Plan of Community Based Health Care (CBHC) for 2017–2022, and Essential Service Package [12,13,14]. The items are the availability of water supply, toilet facility, electricity, thermometer, acute respiratory Infection timer, blood pressure machine, stethoscope, functional height scale, functional weight scale, mid-upper arm circumference scale, and register. For assessing readiness, we considered functional equipment only. 

**The functionality of CGs and CSGs:** To assess the functionality of CG and CSGs, we considered the following four criteria: Availability of the list of CG members.Availability of the list of CSG membersAvailability of the resolution (at the CC) of the CG meeting held within the last month.CC-based service provider reporting about at least one CSG meeting held within the last two months.

### 2.3. Household Survey

#### 2.3.1. Study Design and Study Settings

The household survey was embedded within a larger quasi-experimental study of a project aiming to improve maternal health through the promotion of community participation through CGs and CSGs. The broader baseline study collected data from two intervention upazilas and one comparison upazila. However, the data from two intervention upazilas were relevant to the aim of this paper. The detailed methodology of the larger study is reported elsewhere [15]. In brief, we conducted a community-based, cross-sectional household survey at the study site between March and June 2018. The cross-sectional survey included structured interviews of women with a birth history preceding 12 months of the survey.

#### 2.3.2. Study Population, Sample Size, and Sampling

We adopted stratified cluster sampling. The sampling strategy considered the selected upazilas as the strata and villages (approximately 1000 populations) as clusters. The study adopted probability proportional to size (PPS) sampling to select 20 villages (PPS clusters) from each sub-district. The mapping and listing team developed a map of the selected villages: the maps specified village boundaries, the location of households, and landmarks. Then the mapping team enumerated and listed the households. They identified all women with a recent birth history 12 months preceding the survey. 

The sample size calculation was based on the childbirth attended by a medically trained provider (42% in the Bangladesh Demographic and Health Survey 2014) at the project baseline to detect an expected change of 15 percentage points at the endline at 90% power. The sample size was later adjusted for design effect 2.0 and a 10% non-response rate. The maximum sample size was 416. 

The team invited all eligible respondents for interviews using an interviewer-administered structured questionnaire. The data collection team made three consecutive visits to the households at different times on different days of the week if the women were not available during the first and second visits. Moreover, we recruited data collectors locally to facilitate access to households for data collection. Finally, 454 women from Sarail and 452 women from Kasba were interviewed, with a non-response rate of one percent. Information on the CCs of these two upazilas was also collected. 

#### 2.3.3. Data Collection Tool, Training, and Data Collection

We developed the household survey tool based on the Bangladesh Demographic Health Survey 2014 [16]. Questions related to community engagement and CCs were included to understand women’s knowledge and perception of CCs, CGs, and CSGs. Thirty-three interviewers, locally recruited and trained for data collection, collected data for the household survey. Local recruitment ensured the data collection team’s familiarity with local culture. 

Master trainers of the International Centre for Diarrhoeal Disease Research, Bangladesh (icddr,b) conducted a three-day training for data collectors followed by four days of field practice. The study team pre-tested the data collection tool in non-selected clusters of the study sites and adapted those based on the learnings from the pre-test. Through prior discussions with the women and their family members, interviewers conducted interviews with women alone in the interviewee households. The supervisors ensured data quality by spot-checking data collection, re-interviewing, reviewing filled questionnaires, and providing feedback to data collectors. 

### 2.4. Data Analysis

We analysed the data using Stata 14.0 statistical software (Release 14. College Station, TX, USA, StataCorp LP). We considered accessibility to the CC as ‘good’ if the road condition was reported to be satisfactory in both dry and rainy seasons, ‘moderate’ if the road condition was reported to be satisfactory in the dry season but unsatisfactory in the rainy season, and ‘poor’ if the road condition was reported to be unsatisfactory in both dry and rainy seasons.

Regarding readiness, CCs having 8–10 functional essential items on the day of the visit were considered ‘adequately ready’, whereas CCs with readiness scores of 5–7 were ‘moderately ready’, and CCs having less than five of the essential functional items were regarded as ‘poorly ready’. Regarding functionality, CCs meeting all four functionality criteria were considered well-functioning, whereas CCs failing to meet any of the four criteria were considered poorly functioning. 

The standard steps of principal component analysis were used to generate the socio-economic indices of the households of the interviewed women. Possession of the household, construction material of the house, source of drinking water, type of toilet, land ownership, livestock, etc., were used to generate the index. The wealth quintile was generated based on the indices for the overall sample of households in the study site. Descriptive statistics were used to report on women’s background characteristics, women’s perception regarding CGs, CSGs, and women’s awareness regarding any action within the community from which they benefitted. In addition, separate logistic regression models were performed to assess any significant difference in perception and awareness regarding these actions among women after adjusting for potential confounders and covariates (age, education, religion, family size, parity, involvement in income-generating activities, husband living status, antenatal care (ANC) receiving status and wealth quintile). 

We performed logistic regression to assess the association between the functionality of CGs and CSGs with women’s perception regarding CG and CSGs activities in the community. For these analyses, the functionality scores of CCs were assigned to the surveyed women in their respective CC’s catchment areas. The catchment area was defined by matching the geographic location (villages) of the women’s residence with the geographic location of the CC. Among 906 respondents, we assigned 565 respondents to 54 CCs. We dropped 341 respondents for the logistic regression model as we could not assign them to any CC due to inadequate geographical information in the health facility and household survey data set. Multiple logistic regression models were adopted to control the effect of the covariates and confounders, and associations were presented with adjusted odds ratios (AORs). The odds ratio was adjusted for age, education, and wealth. All ORs and AORs were reported with 95% confidence intervals (CI). 

### 2.5. Ethical Approval and Consent to Participate

Participation in the study was voluntary, and no compensation was provided for participation. The data collection team informed participants about the objectives, study findings’ future use, and associated risks/benefits. If the participant provided written consent to participate, they were recruited for the study. The consent form was prepared in Bengali, the local language. Questionnaires were administered in Bengali. Privacy, anonymity, and confidentiality of participants were strictly maintained. Ethical approval to conduct the study was obtained from the Institutional Review Board of icddr,b (research protocol reference number PR-17088). Permission to conduct the study was also taken from the local health authorities. 

## 3. Findings

### 3.1. A Genealogy of the Community Health Care Project in Bangladesh

Bangladesh was among the signatories to the Alma-Ata Declaration [1,17]. However, even by the late 1990s, national stakeholders recognized that Bangladesh was far from its aim of ensuring Health for All. The lack of availability and accessibility to health care services were identified as two primary reasons for this. The absence of community participation in the area of health care was considered one of the prime gaps [18,19,20]. 

The country adopted the Fourth Population and Health Project (FPHP) in 1992 to reduce fertility and infant mortality and improve maternal and child health. The FPHP was financed by the World Bank, Australia, Canada, Germany, Netherlands, Norway, Sweden, the United Kingdom, and European Union [21]. In 1995, the mid-term review of FPHP took place, and the report stated that the utilisation of public health services was poor [22]. The government, determined to prioritize health services, participated in a Government of Bangladesh (GoB)/donor discussion in Paris in September 1995, which directed the formulation of health sector strategies. Based on the experiences from the FPHP and recommendations from development partners, GoB reformed the health and family planning services and developed the Health and Population Sector Strategy (HPSS) [23]. The Executive Committee of the National Economic Council approved it on 19 August 1997 [24]. Accordingly, the plan document, Health and Population Sector Programme (HPSP) 1998–2003, was developed and operationalized on 1 July 1998. The HPSP 1998–2003 incorporated eight reforms, and two of these were directly linked to the concept of PHC, namely, the ESP and Strengthened Health and Nutrition Services [25]. The government decided to reorganize services by unifying the then separate health and family planning services at three tiers from the upazila level and below to provide ESP; upazila health complex at the upazila level, union health and family welfare centre at the union level, and CCs at the ward or village level [23,25].

Among these three types of facilities, expectations were highest around the CCs, championed by the contemporary prime minister Sheikh Hasina, of the ruling Awami League party to spur the availability of health care. As per the plan, each CC would cover an approximate population of 6000. The CC construction began in 1998, and by 2001, 10,723 CCs out of 13,500 planned were constructed, among which 8000 were functioning between 1998 and 2001 [14,20]. Domiciliary workers of the government, Health Assistants (HAs), and Family Welfare Assistants (FWAs), trained on ESP were the service providers in these CCs [25]. To properly manage the activities of CCs, one CG was formed for each CC. These groups comprised 9–11 members and were led by the land donors, theoretically motivated to contribute to the community’s health of the CC or his representative [18,19,20]. 

Following a regime change and subsequent rule by the Bangladesh Nationalist Party (BNP), all the CCs were closed in 2001 and were not to be reopened until 2008. These establishments were not used as CCs for those seven years and were abandoned. Consequently, over the period, the condition of the infrastructure deteriorated. Some CCs even slipped into rivers due to erosion [18]. 

The 9th national parliament election of Bangladesh took place in December 2008, and the ruling party of 1998, the Awami League, won the majority of seats in the parliament to form the government. This created an opportunity to re-vitalise the CC-based health service delivery model as included in their election manifesto. Therefore, the government started implementing a 5-year long project titled ‘Revitalisation of Community Health Care Initiatives in Bangladesh’ (RCHCIB) in 2009 [19]. Later, the RCHCIB project was included as one of the operational plans titled ‘CBHC, which was implemented under the 3rd Health, Population, and Nutrition Sector Development Programme [26]. Since 2017, the CBHC operational plan has been implemented under the 4th HPNSP 2017–2022 [14]. As of 2019, 13,907 CCs were operational [27]. 

A new cadre of health workers was introduced for RCHCIB, community health care providers (CHCPs), as the primary service provider for CCs. CHCPs are locally recruited and receive 12 weeks of basic training, of which six weeks were allotted for theoretical knowledge and the remaining six weeks for hands-on practical application. Refresher training is to be provided after every two years. The Health Bulletin 2019 reported that 13,907 CHCPs worked in the CCs throughout Bangladesh [9,27]. 

Some modifications were made in the formation and composition of CGs. The number of members was increased from 9–11 to 13–17 to ensure the representation of at least one-third of women and adolescent boys or girls. Provisions were made for the Union Parishad (body of local government) members to head the CGs instead of land donors. According to the new regulations, land donors would be life members and senior vice presidents. The members of the CGs are responsible for confirming the safety, and cleanliness of the CC, addressing problems of a CC using resources (cash and kind) of the community, making people conscious regarding the services provided by CCs, generating funds for proper functioning and maintenance of CCs, inspiring community people to seek care from skilled providers, monitoring the quality of health care activities provided by CCs and supporting the community in accessing quality health care [19,28]. 

In addition to CGs, three CSGs were formed for each CC, comprising 13–17 members with at least one-third female members. To facilitate better coordination, CG representatives were included in each CSG [28]. CSGs were designed to have representation of relevant key stakeholders of the community and people interested in working to improve the health and well-being of the community without any incentive or remuneration [29]. The responsibilities of CSGs included creating awareness among people regarding the nature of services offered at CCs, providing support to people to reach a health facility, advancing health education, generating funds for ensuring emergency services for deprived people, removing misconceptions, and inspiring people to seek care from skilled providers [10,14,28]. 

Among all the responsibilities of CGs and CSGs, one of the key responsibilities is to function as a bridge between the health systems and the community by engaging the community with the health systems to facilitate addressing their own health needs adequately. After forming CGs and CSGs, members were orientated on their roles and responsibilities. The guidelines for these committees recommended that CG members meet monthly and CSG members meet once every two months to discuss the progress made or roles performed, challenges faced, and possible solutions to these [10,20,28]. CHCPs were responsible for sharing the meeting resolutions with respective upazila health managers for their administrative actions and support. These meetings can be considered as one of the critical indicators representing the functionality of the CG-CSG-based community engagement model. Discussion related to activities on awareness-raising, identification of people excluded from health care, needs of the community related to access to health care, etc., are supposed to take place in these meetings. 

Figure 1 depicts the journey of CCs in Bangladesh, which started with the signing of the ‘Alma-Ata Declaration’ in 1978. The journey of CCs has not been linear, but has rather ebbed and flowed, influenced by national and local political agendas. It is therefore perhaps unsurprising that implementation throughout the country is inconsistent and heterogeneous. In the next section, we turn to a case study of CC functioning in the district of Brahmanbaria. 

### 3.2. Community Clinic Accessibility and Readiness

Figure 2 presents the accessibility status of CCs at Kasba and Sarail upazilas. Around 84% of CCs in Kasba had good accessibility. In Sarail, around 33% of CCs had good accessibility, and nearly 48% were moderately accessible.

Figure 3 presents the readiness of the CCs in Kasba and Sarail. Around 30% of CCs in Kasba and 62% of CCs in Sarail had a readiness score of 8 or more, i.e., good readiness. About 21% of CCs at Kasba had a readiness score of 4 or less, i.e., poor readiness. None of the CCs at Sarail had poor readiness. 

Figure 4 presents the functionality of CGs and CSGs at Kasba and Sarail based on the information provided by respective service providers. Around 90% of CCs at Kasba and Sarail could show the member list of their respective CGs and CSGs. Roughly one-third of CCs at Kasba and three-fourths of CCs at Sarail could show the meeting resolutions of the CG meeting of the preceding month. Only about 5% of the CCs at Sarail and Kasba could show the meeting resolutions of the CSG meeting held within the last two months. 

### 3.3. Women’s Perceptions of Community Participation

Table 1 presents the background characteristics of the women who participated in the study. The mean age of the respondents was roughly 25 years (standard deviation ± 4.9). Nearly 50% were aged between 15 and 24 years. Around 20% of respondents had completed less than five years of formal education, while only 23% had completed more than ten years of schooling. Nearly 97% of respondents were Muslims. Very few women (3%) reported being involved in income-generating activities. Roughly 28% of women reported that their husbands were living abroad. Around 68% of women were multipara. About 66% attended antenatal care (ANC) at least once from health facilities. Less than 1% of women attended ANC at a CC. 

Table 2 presents women’s awareness regarding any actions taken by the community to help pregnant women and newborns and perceptions of benefitting from such actions. Only 7% of women overall were aware of any actions taken by the community to help pregnant women and newborns. This was higher among women from Sarail, particularly among those who had more than ten years of education (19%), who were nullipara (14%), and who attended ANC in a health facility (11%). Less than 3% of women reported that they benefited from actions within the community with no apparent variations between Kasba and Sarail and other disaggregating variables.

As presented in Table 3, nearly 21% of women perceived that CGs and CSGs were active in solving problems to improve the health of mothers and newborns. Almost 20% of women perceived that CGs and CSGs were involved in bringing information from the community to health services and vice versa. Thirty-one percent of women perceived that CGs and CSGs were critical in their community. The perception regarding CG and CSG was consistently better among women from Sarail than that of Kasba across all of the categories mentioned above.

Table 4 shows the association between the functionality of CGs and CSGs and women’s perception regarding the role of CGs and CSGs. The odds of perceiving that CGs and CSGs were active in solving MNH problems in the community was 3.2 (CI 2.2–4.6) times higher among women living in the catchment area of a well-functioning CC than that of the poorly-functioning CC. Similar patterns were observed regarding the other perception categories: bringing information from the community back to health services AOR 2.9 (CI 2.0–4.2), CGs and CSGs are important in the community AOR 3.7 (CI 2.6–5.3). 

## 4. Discussion

Being a new nation devastated by nine months of the independence war in 1971, Bangladesh faced significant internal challenges in restructuring its public service sector, including the health sector. One of the major challenges was the military and quasi-military rule that continued for 15 years between 1975 and 1990. For this period, the emphasis was on establishing health facilities, increasing and strengthening the health workforce, reinforcing population control programmes, and involvement of NGOs and private health facilities [30,31]. Hence the three health plans implemented during this period did not adequately reflect the core components of PHC, particularly community participation [31]. The political journey of Bangladesh partially explains why it took around two decades, till the mid 90′s, to establish CCs as a focused approach for promoting community participation in PHC [18]. 

The decision to introduce CCs is engraved in the context and the initiatives are taken then. Since 1990, the external core donors, including the World Bank, wanted to ensure equity and efficiency in allocating resources in the health sector and recommended a great institutional reform in the health sector. In 1996, the World Bank and other development partners pushed the government to adopt a comprehensive and sector-wide approach (SWAp) instead of project-based health service delivery [17]. Hence, the SWAp replaced 128 projects in 1998 to deliver more consolidated and better-planned health services nationwide [21]. The CC initiative and the adoption of the SWAp in Bangladesh share the same timeline, which also potentially explains the context and motivation of the policymakers during their inception phase [8,17,21]. Later, the CC initiative was integrated within the SWAp as a dedicated operational plan named CBHC [9]. 

The journey of CCs is also a reflection of the political ideologies, agendas, and priorities of the Awami League and BNP, the two main political parties in Bangladesh. While ideologically Awami League is more attached to socialism, BNP tends to be more oriented towards open markets and capitalism [32,33]. The rivalry between these two parties animated the Bangladeshi political landscape since 1991 when the BNP formed a democratically-elected government after winning the majority of seats against the Awami League and other political parties after the fall of military rule under General Ershad [34,35]. It would be difficult to find issues on which both parties came to a consensus, let alone take on an unfinished task of the opposition following a regime change [36]. In this sense, the pendulum of support for the CC initiative since its introduction must be read within the political contest between Awami League and BNP. 

This political tug-of-war may at least partly explain the compromised CC readiness we found in our study, as well as accessibility, functionality, and perception of the community regarding the role of CG, CSG members, and their performance. This is not a stand-alone finding. The BHFS–2017 revealed key readiness gaps of CCs across the country. Additionally, the findings shared in the independent evaluation report of the WHO is also comparable with our study findings. Like our study, the independent evaluation report mentioned poor road conditions during the rainy season as a barrier to accessing the CCs [20]. The readiness score of the CCs assessed in our study area was low; 62% of the CCs of Sarail and 30% of the CCs of Kasba had eight or more basic equipment. This score was better than CCs reported in the Bangladesh Health Facility Survey 2017-18; only 22.8% of CCs had six basic equipment (adult scale, child scale, thermometer, stethoscope, blood pressure apparatus, light source) [37]. To make our assessment tool more comprehensive, we also reviewed the Operational Plan of the 4th HPNSP and Essential Service Package beside the BHFS-2014 tool. In addition, bureaucratic complexities, prevalent in most of the public sectors of Bangladesh, including the health sector, could also be an explanatory factor. The Bangladesh health system has often suffered due to the complex procurement and supply chain system [38]. This may also explain some parts of the readiness status of these CCs. 

According to the findings of our study, most of the CCs’ community participation mechanism was not functioning well. CSGs were less functional than CGs. Some of the CCs could not even provide the list of CG and CSG members. This finding aligns with findings reported by the Independent Evaluation of Community-Based Health Services in Bangladesh. The report shared that it was more challenging to ensure the organization of CSG meetings than CG meetings. It also mentioned some factors which they thought had been responsible for this poor functionality; CSG members’ lack of understanding regarding their roles as CSG members, lack of ownership among them, their other priorities, and expectations of CG, CSG members for refreshments. In addition to these, it also mentioned weak monitoring and leadership in the health systems as another contributing factor to CSGs’ poor functionality [20]. 

To further explain the poor functionality of CG-CSGs, the CG-CSG model of community participation could also be critiqued. Firstly, the CG-CSG model of community participation has an inherent issue with a power imbalance. The member of the Union Parishad is the chair of the CC by position, and the landowner is the vice president [10]. Both are potent members of the community. Therefore, rather than unsettling existing power dynamics, the CG-CSG mirrors these imbalances, compromising meaningful engagement of community and disadvantaged populations. The potential imbalance in power dynamics can make the proposed participation of women, adolescents, and poor people tokenistic. Secondly, the CCs are established on land donated by the community. Commonly, the land donor is a relatively wealthy individual, often a local leader affiliated with the political party of the incumbent government. The donated land tends to be located in inaccessible areas of the village or strategically positioned to obtain a political advantage in the community. Hence accessibility and the participation of the broader community in facility management are sometimes challenging. Thirdly, the CGs are expected to take the management responsibilities of their respective CCs and generate funds to address the gaps related to health services in their community. However, there is no allocation from the government to support the function of CG-CSGs except for their training [9]. This is very different from the other community participation models that exist in Bangladesh. For instance, the Union Parishad and Upazila Parishad receive direct funds from the Ministry of Local Government, Rural Development & Co-operatives to implement their plans and recommendations at a local level [14,39]. In addition, CG-CSGs members work on a volunteer basis. This raised ethical implications and calls into question the motivations of participants, given that some members are identified as representing the most disadvantaged in the community and are being asked to work without remuneration. Hence, the issues with functionality that we have identified regarding the CCs can be read as flowing from their design. 

Finally, it is essential to understand the different expectations of the community regarding community participation as well as the support that they need and expect from the CGs and CSGs. Disconnects between the expectations of the community and the expectations of the government and policymakers who are in charge of navigating this community participation model can complicate moving beyond tokenistic community participation.

Most importantly, according to our findings, the perception of the community women regarding the role of CG, CSG members, and CG-CSGs’ functionality was poor in general. However, it was significantly better when the CG-CSG members met regularly. Regular meetings of CG, and CSG indicate that the CG-CSG members are active in the community and sincerely undertake their assigned responsibilities. 

## 5. Limitations

The CC assessments were conducted five months before the household survey in March 2018. The readiness and functionality of CCs are highly unlikely to be changed substantially during this period. We acknowledge that women’s responses to CCs, CGs, and CSGs could be different if the household survey were conducted simultaneously with CC assessments. This study assessed the CCs’ accessibility status based on the condition of the access road to the CCs in both dry and rainy seasons, as reported by CC-based healthcare providers. Hence, there is a slim chance of recall bias and reporting error. Another limitation of the study is its cross-sectional design. Therefore, it could not establish causality between the functionality of the CC-based community participation model and women’s perception, knowledge, and awareness related to CGs and CSGs. However, the authors tried to interpret the findings cautiously and presented the association after necessary adjustments. Another noteworthy limitation is, that due to the unavailability of defined catchment geography of some CCs, some of the participants’ locations could not be matched with CCs’ locations. Hence, those participants were excluded from the corresponding analysis. However, we compared the demographic characteristics of those who were included and those who were not included, and we found no significant difference.

## 6. Conclusions

For women in the rural community in Bangladesh, the CCs are the first point of contact with the public health facility. However, there are gaps in the CCs’ readiness and functionality, which are primarily induced by the shortcoming of the CG-CSG-based community participation model. Proper understanding is needed to address this problem which has many facets and layers, including political priorities, expectations, and provisions at a local level. An improved understanding will help design a better community participation model and use it for improving maternal health. 

## Figures and Tables

**Figure 1 ijerph-20-02271-f001:**
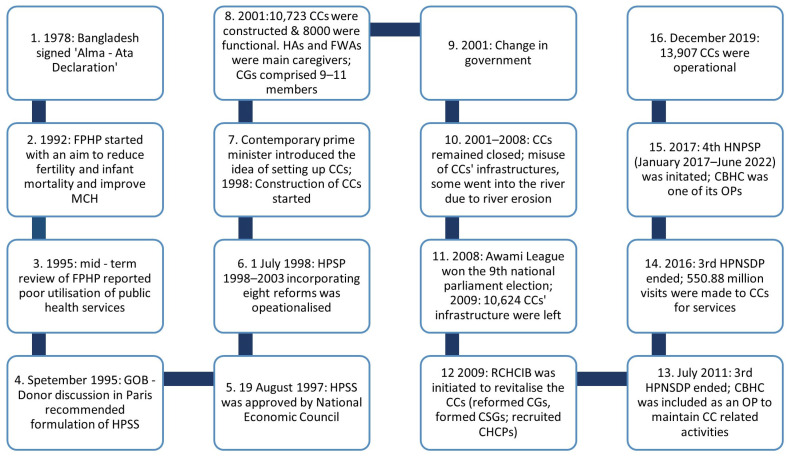
Timeline of the CC journey in Bangladesh.

**Figure 2 ijerph-20-02271-f002:**
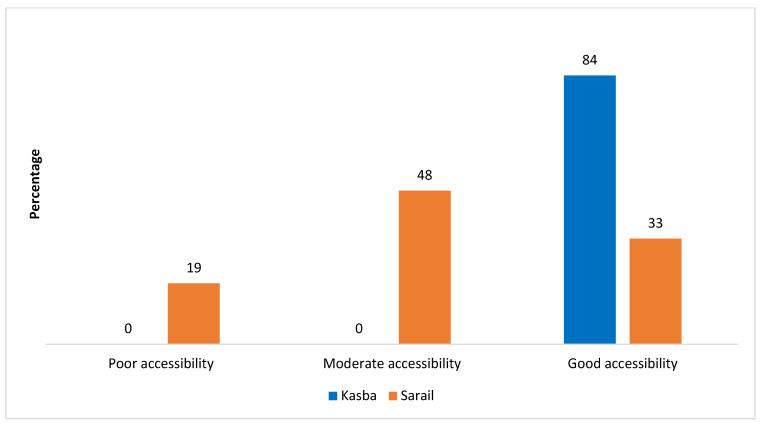
Accessibility status of CCs at Kasba (n = 33) and Sarail (n = 21). Data from 5/33 CCs were missing (staff absent in four CCs and one CC was destroyed).

**Figure 3 ijerph-20-02271-f003:**
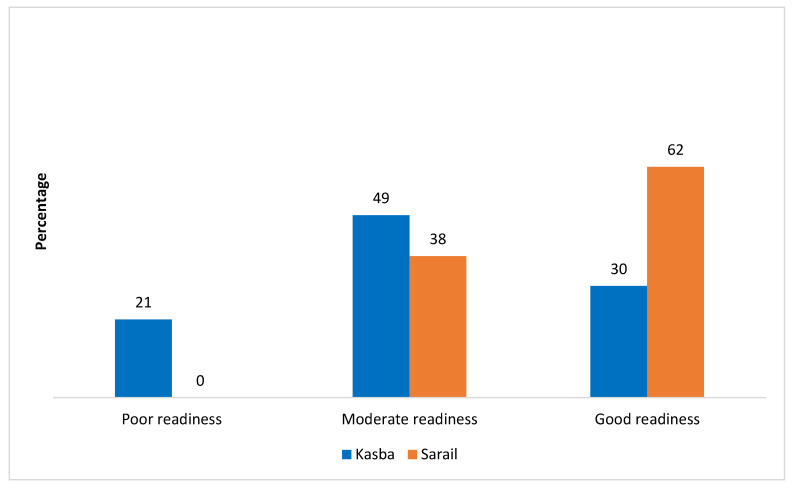
Readiness status of CCs at Kasba (n = 33) and Sarail (n = 21).

**Figure 4 ijerph-20-02271-f004:**
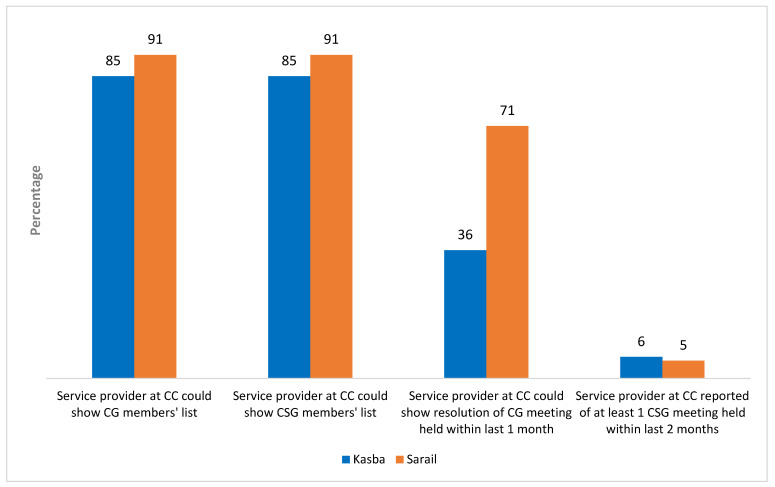
Functionality of CGs and CSGs at Kasba (n = 33) and Sarail (n = 21).

**Table 1 ijerph-20-02271-t001:** Background characteristics of respondents (Kasba = 452 and Sarail = 454).

Background Characteristics	Kasba (N = 452)	Sarail (N = 454)	Total (N = 906)
	% (n)	% (n)	% (n)
**Age**			
15–24 years	49.8 (225)	50 (227)	49.9 (452)
25–34 years	46 (208)	41.4 (188)	43.7 (396)
35+ years	4.2 (19)	8.6 (39)	6.4 (58)
Mean age in years (SD)	24.9 (±4.8)	25.1 (±5.1)	25.0 (±4.9)
**Education/Schooling**			
0–4 years	11.9 (54)	28.6 (130)	20.3 (184)
5–10 years	59.1 (267)	55.1 (250)	57.1 (517)
10+ years	29 (131)	16.3 (74)	22.6 (205)
Mean years of schooling (SD)	7.7 (±3.2)	5.7 (±3.4)	6.7 (±3.5)
**Religion**			
Muslim	95.8 (433)	97.8 (444)	96.8 (877)
Others	4.2 (19)	2.2 (10)	3.2 (29)
**Family size**			
1–4	21.9 (99)	26.2 (119)	24.1 (218)
5 or more	78.1 (353)	73.8 (335)	75.9 (688)
**Parity ***			
Nullipara	35.4 (160)	27.5 (125)	31.5 (285)
Multipara	64.4 (291)	72 (327)	68.2 (618)
**Involved in income-generating activities**			
Yes	2.2 (10)	3.7 (17)	3 (27)
No	97.8 (442)	96.3 (437)	97 (879)
**Living status of husband**			
Lives with wife	41.4 (187)	62.1 (282)	51.8 (469)
Lives in other places but within country	23.2 (105)	16.7 (76)	20 (181)
Lives abroad	35.4 (160)	21.1 (96)	28.3 (256)
**Attended ANC in a health facility**			
Yes	67.9 (307)	65 (295)	66.4 (602)
No	32.1 (145)	35 (159)	33.6 (304)
**Attended ANC in a CC**			
Yes	0.9 (4)	0.4 (2)	0.7 (6)
No	99.1 (448)	99.6 (452)	99.3 (900)
**Wealth quintile**			
Lowest	8 (36)	30.6 (139)	19.3 (175)
Second	17.9 (81)	22.2 (101)	20.1 (182)
Middle	22.6 (102)	19.6 (89)	21.1 (191)
Fourth	25 (113)	13.9 (63)	19.4 (176)
Highest	26.5 (120)	13.7 (62)	20.1 (182)

* Three responses were missing.

**Table 2 ijerph-20-02271-t002:** Women’s awareness regarding any actions within the community *, and got themselves benefitted from any actions within the community (Kasba = 452 and Sarail = 454).

Background Characteristics	Women Aware of Any Actions within the Community Taken to Help Pregnant Women and Newborns			Women Themselves Got Benefitted from Any Actions within the Community Taken to Help Pregnant Women and Newborns		
	Kasba	Sarail	Total	Kasba	Sarail	Total
	% (n)	% (n)	% (n)	% (n)	% (n)	% (n)
Total	5.5 (25)	8.4 (38)	7.0 (63)	2.4 (11)	2.4 (11)	2.4 (22)
**Age**						
15–24 years	3.6 (8)	10.6 (24)	7.1 (32)	1.3 (3)	2.6 (6)	2.0 (9)
25–34 years	7.7 (16)	6.9 (13)	7.3 (29)	3.8 (8)	2.1 (4)	3.0 (12)
35+ years	5.3 (1)	2.6 (1)	3.4 (2)	0 (0)	2.6 (1)	1.7 (1)
** *p* ** **value**	0.170	0.160	0.550	0.186	0.942	0.579
**Education/Schooling**						
0–4 years	1.9 (1)	3.8 (5)	3.3 (6)	0 (0)	1.5 (2)	1.1 (2)
5–10 years	6.4 (17)	7.6 (19)	7.0 (36)	3 (8)	2.4 (6)	2.7 (14)
10+ years	5.3 (7)	18.9 (14)	10.2 (21)	2.3 (3)	4.1 (3)	2.9 (6)
** *p* ** **-value**	0.414	0.001	0.026	0.424	0.532	0.410
**Religion**						
Muslim	5.3 (23)	8.6 (38)	7.0 (61)	2.5 (11)	2.5 (11)	2.5 (22)
Others (Hindu/Christian etc.)	10.5 (2)	0 (0)	6.9 (2)	0 (0)	0 (0)	0.0 (0)
** *p* ** **-value**	0.330	0.334	0.990	0.482	0.614	0.388
**Family size**						
1–4	4 (4)	7.6 (9)	6.0 (13)	2 (2)	2.5 (3)	2.3 (5)
5 or more	5.9 (21)	8.7 (29)	7.3 (50)	2.5 (9)	2.4 (8)	2.5 (17)
** *p* ** **-value**	0.463	0.711	0.509	0.763	0.935	0.882
**Involved in income-generating activities**						
Yes	10 (1)	11.8 (2)	6.8 (60)	10 (1)	0 (0)	2.4 (21)
No	5.4 (24)	8.2 (36)	11.1 (3)	2.3 (10)	2.5 (11)	3.7 (1)
** *p* ** **-value**	0.532	0.606	0.389	0.116	0.508	0.662
**Living status of husband**						
Lives with wife	4.8 (9)	8.5 (24)	7 (33)	1.1 (2)	2.5 (7)	1.9 (9)
Lives in other places but within country	4.8 (5)	7.9 (6)	6.1 (11)	1.9 (2)	2.6 (2)	2.2 (4)
Lives abroad	6.9 (11)	8.3 (8)	7.4 (19)	4.4 (7)	2.1 (2)	3.5 (9)
** *p* ** **-value**	0.652	0.985	0.858	0.127	0.968	0.401
**Parity**						
Nullipara	5 (8)	13.6 (17)	8.8 (25)	3.1 (5)	3.2 (4)	3.2 (9)
Multipara	5.8 (17)	6.4 (21)	6.1 (38)	2.1 (6)	2.1 (7)	2.1 (13)
**Attended ANC in a health facility**						
Yes	5.9 (18)	10.5 (31)	8.1 (49)	2 (6)	3.4 (10)	2.7 (16)
No	4.8 (7)	4.4 (7)	4.6 (14)	3.4 (5)	0.6 (1)	2 (6)
** *p* ** **-value**	0.653	0.025	0.048	0.336	0.068	0.528
**Attended ANC in a CC**						
Yes	25 (1)	50 (1)	33.3 (2)	0 (0)	0 (0)	0 (0)
No	5.4 (24)	8.2 (37)	6.8 (61)	2.5 (11)	2.4 (11)	2.4 (22)
** *p* ** **-value**	0.087	0.033	0.011	0.751	0.823	0.698
**Wealth quintile**						
Lowest	11.1 (4)	5 (7)	6.3 (11)	2.8 (1)	0.7 (1)	1.1 (2)
Second	2.5 (2)	6.9 (7)	4.9 (9)	1.2 (1)	5 (5)	3.3 (6)
Middle	4.9 (5)	10.1 (9)	7.3 (14)	2.9 (3)	1.1 (1)	2.1 (4)
Fourth	6.2 (7)	11.1 (7)	8.0 (14)	2.7 (3)	1.6 (1)	2.3 (4)
Highest	5.8 (7)	12.9 (8)	8.2 (15)	2.5 (3)	4.8 (3)	3.3 (6)
** *p* ** **-value**	0.435	0.296	0.729	0.958	0.148	0.647

* Actions with the community include; community education/sensitization on MNH issues, community-organized transportation to reach MNH services, community-organized funds to assist women in paying for MNH services, community involvement in planning/providing feedback to health services, etc.

**Table 3 ijerph-20-02271-t003:** Women’s perception regarding CGs and CSGs (Kasba = 452 and Sarail = 454).

Background Characteristics	The CGs/CSGs Are Active in Solving Problems to Improve the Health of Mothers and Newborns in Your Community			The CGs/CSGs Are Active in Bringing Information from the Community Back to the Health Services			The CGs/CSGs Are Active in Bringing Information from the Health Services to the Community			The CGs/CSGs Are Important in Your Community		
	Kasba	Sarail	Total	Kasba	Sarail	Total	Kasba	Sarail	Total	Kasba	Sarail	Total
	% (n)	% (n)	% (n)	% (n)	% (n)	% (n)	% (n)	% (n)	% (n)	% (n)	% (n)	% (n)
**Age**												
15–24 years	13.3 (30)	36.1 (82)	24.8 (112)	12.4 (28)	33 (75)	22.8 (103)	12.4 (28)	31.3 (71)	21.9 (99)	28.4 (64)	37.4 (85)	33 (149)
25–34 years	15.4 (32)	19.1 (36)	17.2 (68)	14.9 (31)	18.1 (34)	16.4 (65)	16.8 (35)	17.6 (33)	17.2 (68)	31.7 (66)	27.7 (52)	29.8 (118)
35+ years	26.3 (5)	7.7 (3)	13.8 (8)	26.3 (5)	5.1 (2)	12.1 (7)	26.3 (5)	2.6 (1)	10.3 (6)	31.6 (6)	10.3 (4)	17.2 (10)
** *p* ** **-value**	0.296	0.000	0.010	0.229	0.000	0.022	0.166	0.000	0.047	0.750	0.001	0.045
**Education/Schooling**												
0–4 years	14.8 (8)	13.1 (17)	13.6 (25)	13 (7)	12.3 (16)	12.5 (23)	13 (7)	12.3 (16)	12.5 (23)	25.9 (14)	19.2 (25)	21.2 (39)
5–10 years	16.5 (44)	28.8 (72)	22.4 (116)	15.7 (42)	26.4 (66)	20.9 (108)	16.9 (45)	24.4 (61)	20.5 (106)	29.2 (78)	34.4 (86)	31.7 (164)
10+ years	11.5 (15)	43.2 (32)	22.9 (47)	11.5 (15)	39.2 (29)	21.5 (44)	12.2 (16)	37.8 (28)	21.5 (44)	33.6 (44)	40.5 (30)	36.1 (74)
** *p* ** **-value**	0.415	0.000	0.027	0.497	0.000	0.032	0.430	0.000	0.037	0.521	0.002	0.004
**Religion**												
Muslim	14.5 (63)	27 (120)	20.9 (183)	14.1 (61)	24.8 (110)	19.5 (171)	15 (65)	23.4 (104)	19.3 (169)	30.5 (132)	31.5 (140)	31 (272)
Others	21.1 (4)	10 (1)	17.2 (5)	15.8 (3)	10 (1)	13.8 (4)	15.8 (3)	10 (1)	13.8 (4)	21.1 (4)	10 (1)	17.2 (5)
** *p* ** **-value**	0.435	0.228	0.636	0.835	0.282	0.444	0.926	0.319	0.460	0.380	0.146	0.113
**Family size**												
1–4	15.2 (15)	27.7 (33)	22 (48)	14.1 (14)	25.2 (30)	20.2 (44)	15.2 (15)	21.8 (26)	18.8 (41)	35.4 (35)	31.1 (37)	33 (72)
5 or more	14.7 (52)	26.3 (88)	20.3 (140)	14.2 (50)	24.2 (81)	19 (131)	15 (53)	23.6 (79)	19.2 (132)	28.6 (101)	31 (104)	29.8 (205)
** *p* ** **-value**	0.917	0.757	0.596	0.995	0.822	0.710	0.973	0.700	0.901	0.196	0.992	0.367
**Parity**												
Nullipara	20 (32)	33.6 (42)	26 (74)	18.8 (30)	30.4 (38)	23.9 (68)	20.6 (33)	28 (35)	23.9 (68)	36.3 (58)	33.6 (42)	35.1 (100)
Multipara	12 (35)	23.5 (77)	18.1 (112)	11.7 (34)	21.7 (71)	17 (105)	12 (35)	20.8 (68)	16.7 (103)	26.5 (77)	30.3 (99)	28.5 (176)
** *p* ** **-value**	0.023	0.030	0.007	0.040	0.053	0.015	0.015	0.102	0.010	0.030	0.495	0.045
**Involved in income-generating activities**												
Yes	0 (0)	29.4 (5)	18.5 (5)	0 (0)	23.5 (4)	14.8 (4)	0 (0)	11.8 (2)	7.4 (2)	20 (2)	23.5 (4)	22.2 (6)
No	15.2 (67)	26.5 (116)	20.8 (183)	14.5 (64)	24.5 (107)	19.5 (171)	15.4 (68)	23.6 (103)	19.5 (171)	30.3 (134)	31.4 (137)	30.8 (271)
** *p* ** **-value**	0.182	0.793	0.772	0.194	0.928	0.548	0.178	0.257	0.117	0.482	0.494	0.339
**Living status of husband**												
Lives with wife	11.8 (22)	32.3 (91)	24.1 (113)	11.2 (21)	29.1 (82)	22 (103)	12.3 (23)	28.7 (81)	22.2 (104)	32.6 (61)	35.8 (101)	34.5 (162)
Lives in other places but within country	25.7 (27)	7.9 (6)	18.2 (33)	24.8 (26)	7.9 (6)	17.7 (32)	24.8 (26)	6.6 (5)	17.1 (31)	36.2 (38)	15.8 (12)	27.6 (50)
Lives abroad	11.3 (18)	25 (24)	16.4 (42)	10.6 (17)	24 (23)	15.6 (40)	11.9 (19)	19.8 (19)	14.8 (38)	23.1 (37)	29.2 (28)	25.4 (65)
** *p* ** **-value**	0.002	0.000	0.033	0.002	0.001	0.098	0.006	0.000	0.042	0.047	0.003	0.024
**Attended ANC in a health facility**												
Yes	10.3 (15)	20.8 (33)	15.8 (48)	8.3 (12)	18.2 (29)	13.5 (41)	9 (13)	17.6 (28)	13.5 (41)	19.3 (28)	25.8 (41)	22.7 (69)
No	16.9 (52)	29.8 (88)	23.3 (140)	16.9 (52)	27.8 (82)	22.3 (134)	17.9 (55)	26.1 (77)	21.9 (132)	35.2 (108)	33.9 (100)	34.6 (208)
** *p* ** **-value**	0.066	0.037	0.009	0.014	0.024	0.002	0.013	0.041	0.002	0.001	0.075	0.000
**Attended ANC in a CC**												
Yes	25 (1)	50 (1)	33.3 (2)	25 (1)	100 (2)	50 (3)	0 (0)	50 (1)	16.7 (1)	50 (2)	50 (1)	50 (3)
No	14.7 (66)	26.5 (120)	20.7 (186)	14.1 (63)	24.1 (109)	19.1 (172)	15.2 (68)	23 (104)	19.1 (172)	29.9 (134)	31 (140)	30.4 (274)
** *p* ** **-value**	0.565	0.454	0.446	0.532	0.013	0.056	0.398	0.366	0.879	0.383	0.562	0.300
**Wealth quintile**												
Lowest	16.7 (6)	12.9 (18)	13.7 (24)	19.4 (7)	12.9 (18)	14.3 (25)	19.4 (7)	11.5 (16)	13.1 (23)	30.6 (11)	15.8 (22)	18.9 (33)
Second	13.6 (11)	30.7 (31)	23.1 (42)	9.9 (8)	27.7 (28)	19.8 (36)	12.3 (10)	25.7 (26)	19.8 (36)	25.9 (21)	36.6 (37)	31.9 (58)
Middle	15.7 (16)	29.2 (26)	22 (42)	14.7 (15)	25.8 (23)	19.9 (38)	15.7 (16)	24.7 (22)	19.9 (38)	33.3 (34)	36 (32)	34.6 (66)
Fourth	11.5 (13)	41.3 (26)	22.2 (39)	12.4 (14)	36.5 (23)	21 (37)	13.3 (15)	36.5 (23)	21.6 (38)	23.9 (27)	39.7 (25)	29.5 (52)
Highest	17.5 (21)	32.3 (20)	22.5 (41)	16.7 (20)	30.6 (19)	21.4 (39)	16.7 (20)	29 (18)	20.9 (38)	35.8 (43)	40.3 (25)	37.4 (68)
** *p* ** **-value**	0.750	0.000	0.158	0.560	0.002	0.440	0.815	0.001	0.264	0.275	0.000	0.002
**Total**	14.8 (67)	26.7 (121)	20.8 (188)	14.2 (64)	24.4 (111)	19.3 (175)	15 (68)	23.1 (105)	19.1 (173)	30.1 (136)	31.1 (141)	30.6 (277)

**Table 4 ijerph-20-02271-t004:** Association between the functionality of CGs and CSGs and women’s perception of the role of CGs and CSGs; (N = 565).

Functionality of CGs and CSGs	N	%	The CGs/CSGs Are Active in Solving Problems to Improve the Health of Mothers and Newborns in Your Community			The CGs/CSGs Are Active in Bringing Information from the Community Back to the Health Services			The CGs/CSGs Are Active in Bringing Information from the Health Services to the Community			The CGs/CSGs Are Important in Your Community		
			%	OR	AOR	%	OR	AOR	%	OR	AOR	%	OR	AOR
**Poorly-Functioning ****	305	54.0	20.7	Ref	Ref	19.3	Ref	Ref	18.7	Ref	Ref	25.2	Ref	Ref
**Well-Functioning *****	260	46.0	45.4	3.2 (2.2, 4.6)	3.2 (2.2, 4.6)	41.2	2.9 (2.0, 4.2)	2.9 (2.0, 4.2)	41.9	3.1 (2.2, 4.6)	3.2 (2.2, 4.6)	54.6	3.6 (2.5, 5.1)	3.7 (2.6, 5.3)

We adjusted the odds ratio for age, education and wealth. ** CC was either closed or infrastructure was broken, or CC-based healthcare providers could not show the list of CG, and CSG members. *** CC-based health care provider could show the list of CG, CSG members, resolution of CG meeting held within the last month, and could inform about at least one CSG meeting held within the previous two months.

## Data Availability

The datasets will be available upon a valid request to the corresponding author (Ahmed Ehsanur Rahman, ehsanur@icddrb.org).

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
