# Peer review of "Taking a Pulse on Community Participation in Maternal Health through Community Clinics in Bangladesh"

_ijerph, 2023, doi:10.3390/ijerph20032271_

Round 1
Reviewer 1 Report
This article evaluates 3 characteristics of community centers: accessibility, preparation, and functionality (based on several items), but in abstract the authors only mention functionality as if this characteristic is more important than accessibility and preparation, perhaps this is according to the number of words. You can improve this section.
Please better edit Table 3.
Line 135: Define the meaning of icddr,b (I assume it is, from line 10, International Center for Diarrhoeal Disease Research, Bangladesh), but You must define it in the body of the text.
Line 165 - Define the meaning of ANC (I assume it is Antennanatal Care), but you must define it in the body of the text.
Lines 113-117: First, how do You calculate the sample size of 906 (454 + 452) women? What were the criteria for selecting this sample size? Second, lines 171-173, it is not clear why there are respondents who did not match with CCs since you developed the household survey tool. Third when You dropped 341 respondents, your analysis may be biased or underpowered to run a logistic regression model, but in lines 462-465, You did not mention this.
Lines 331-335: In this section there is a different font, in lines 334-335 the table number is wrong, plus it dropped 341 respondents, Kasba and Sarail are no longer equal to 452 and 454, respectively.
Figures 2, 3, 4: It might be useful to design these figures with two Y axes, a left axis with percentages and a right axis with frequencies.
Figure 2: Is a bar missing for Kasba percentage? In Sarail you can add 19+48+33=100, but for Kasba there is only one bar.
Line 346: Why did you adjust the logistic regression models only for age, education, and wealth, if you have other significant variables in Table 3, for example, "husband's living status" or "attended prenatal care at a of health" or "parity"?
Reviewer 2 Report
You have worked hard work in doing this manuscript. Unfortunately this manuscript only suggest the general characteristics of the subject and the good or bad perception of CG and CSG, and the significance of this manuscript to the academic world seems to be very insufficient.
It would be more meaningful if it was a study that evaluated the effectiveness of CG and CSG.
The content of the survey is too simplistic.
You said that multiple logistic regression analysis was performed to determine the association between the function and perception of the good or bad perception of CG and CSG, but it seems to be an error.
I suggest some review comments.
